# On preserving non-discrimination when combining expert advice

**Avrim Blum**
TTI-Chicago
avrim@ttic.edu

**Suriya Gunasekar**
TTI-Chicago
suriya@ttic.edu

**Thodoris Lykouris**
Cornell University
teddlyk@cs.cornell.edu

**Nathan Srebro**
TTI-Chicago
nati@ttic.edu

## Abstract

We study the interplay between sequential decision making and avoiding discrimination against protected groups, when examples arrive online and do not follow distributional assumptions. We consider the most basic extension of classical online learning: *Given a class of predictors that are individually non-discriminatory with respect to a particular metric, how can we combine them to perform as well as the best predictor, while preserving non-discrimination?* Surprisingly we show that this task is unachievable for the prevalent notion of *equalized odds* that requires equal false negative rates and equal false positive rates across groups. On the positive side, for another notion of non-discrimination, *equalized error rates*, we show that running separate instances of the classical multiplicative weights algorithm for each group achieves this guarantee. Interestingly, even for this notion, we show that algorithms with stronger performance guarantees than multiplicative weights cannot preserve non-discrimination.

## 1 Introduction

The emergence of machine learning in the last decade has given rise to an important debate regarding the ethical and societal responsibility of its offspring. Machine learning has provided a universal toolbox enhancing the decision making in many disciplines from advertising and recommender systems to education and criminal justice. Unfortunately, both the data and their processing can be biased against specific population groups (even inadvertently) in every single step of the process [4]. This has generated societal and policy interest in understanding the sources of this discrimination and interdisciplinary research has attempted to mitigate its shortcomings.

Discrimination is commonly an issue in applications where decisions need to be made sequentially. The most prominent such application is online advertising where platforms need to sequentially select which ad to display in response to particular query searches. This process can introduce discrimination against protected groups in many ways such as filtering particular alternatives [12, 2] and reinforcing existing stereotypes through search results [38, 25]. Another canonical example of sequential decision making is medical trials where underexploration on female groups often leads to significantly worse treatments for them [31]. Similar issues occur in image classification as stressed by "gender shades" [7]. The reverse (overexploration in minority populations) can also cause concerns especially if conducted in a non-transparent fashion [5].

In these sequential settings, the assumption that data are i.i.d. is often violated. Online advertising, recommender systems, medical trials, image classification, loan decisions, criminal recidivism all require decisions to be made sequentially. The corresponding labels are not identical across time and can be affected by the economy, recent events, etc. Similarly labels are also not independent across rounds – if a bank offers a loan then this decision can affect whether the loanee or their environment will be able to repay future loans thereby affecting future labels as discussed by Liu et al. [32]. As a result, it is important to understand the effect of this adaptivity on non-discrimination.

The classical way to model settings that are not i.i.d. is via adversarial online learning [30, 17], which poses the question: *Given a class $\mathcal{F}$ of predictors, how can we make online predictions that perform as well as the best predictor from $\mathcal{F}$ in hindsight?* The most basic online learning question (answered via the celebrated "multiplicative weights" algorithm) concerns competing with a finite set of predictors. The class $\mathcal{F}$ is typically referred to as "experts" and can be thought as "features" of the example where we want to make online predictions that compete with the best 1-sparse predictor.

In this work, we wish to understand the interplay between adaptivity and non-discrimination and therefore consider the most basic extension of the classical online learning question:

> *Given a class of **individually non-discriminatory predictors**, how can we combine them to perform as well as the best predictor, while preserving non-discrimination?*

The assumption that predictors are individually non-discriminatory is a strong assumption on the predictors and makes the task trivial in the batch setting where the algorithm is given labeled examples and wishes to perform well on unseen examples drawn from the same distribution. This happens because the algorithm can learn the best predictor from the labeled examples and then follow it (since this predictor is individually non-discriminatory, the algorithm does not exhibit discrimination). This enables us to understand the potential overhead that adaptivity is causing and significantly strengthens any impossibility result. Moreover, we can assume that predictors have been individually vetted to satisfy the non-discrimination desiderata – we therefore wish to understand how to efficiently compose these non-discriminatory predictors while preserving non-discrimination.

## 1.1   Our contribution

**Our impossibility results for equalized odds.**   Surprisingly, we show that for a prevalent notion of non-discrimination, *equalized odds*, it is impossible to preserve non-discrimination while also competing comparably to the best predictor in hindsight (no-regret property). Equalized odds, suggested by Hardt et al. [20] in the batch setting, restricts the set of allowed predictors requiring that, when examples come from different groups, the prediction is independent to the group conditioned on the label. In binary classification, this means that the false negative rate (fraction of positive examples predicted negative) is equal across groups and the same holds for the false positive rate (defined analogously). This notion was popularized by a recent debate on potential bias of machine learning risk tools for criminal recidivism [1, 10, 28, 16].

Our impossibility results demonstrate that the order in which examples arrive significantly complicates the task of achieving desired efficiency while preserving non-discrimination with respect to equalized odds. In particular, we show that any algorithm agnostic to the group identity either cannot achieve performance comparable to the best predictor or exhibits discrimination in some instances (Theorem 1). This occurs in phenomenally simple settings with only two individually non-discriminatory predictors, two groups, and perfectly balanced instances: groups are of equal size and each receives equal number of positive and negative labels. The only imbalance exists in the order in which the labels arrive which is also relatively well behaved – labels are generated from two i.i.d. distributions, one in the first half of the instance and one in the second half. Although in many settings we cannot actively use the group identity of the examples due to legal reasons (e.g., in hiring), one may wonder whether these impossibility results disappear if we can actively use the group information to compensate for past mistakes. We show that this is also not the case (Theorem 2). Although our groups are not perfectly balanced, the construction is again very simple and consists only of two groups and two predictors: one always predicting positive and one always predicting negative. The simplicity of the settings, combined with the very strong assumption on the predictors being individually non-discriminatory speaks to the trade-off between adaptivity and non-discrimination with respect to equalized odds.

**Our results for equalized error rates.**   The strong impossibility results with respect to equalized odds invite the natural question of whether there exists some alternative fairness notion that, given access to non-discriminatory predictors, achieves efficiency while preserving non-discrimination. We answer the above positively by suggesting the notion of *equalized error rates*, which requires that the average expected loss (regardless whether it stems from false positives or false negatives) encountered by each group should be the same. This notion makes sense in settings where performance and fairness are measured with respect to the same objective. Consider a medical application where people

from different subpopulations wish to receive appropriate treatment and any error in treatment costs equally both towards performance and towards fairness.[1] It is morally objectionable to discriminate against one group, e.g. based on race, using it as experimentation to enhance the quality of service of the other, and it is reasonable to require that all subpopulations receive same quality of service.

For this notion, we show that, if all predictors are individually non-discriminatory with respect to equalized error rates, running separate multiplicative weights algorithms, one for each subpopulation, preserves this non-discrimination without decay in the efficiency (Theorem 3). The key property we use is that the multiplicative weights algorithm guarantees to perform not only no worse than the best predictor in hindsight but also no better; this property holds for a broader class of algorithms [14]. Our result applies to general loss functions beyond binary predictions and only requires predictors to satisfy the weakened assumption of being approximately non-discriminatory.

Finally, we examine whether the decisions of running separate algorithms and running this particular not so efficient algorithm were important for the result. We first give evidence that running separate algorithms is essential for the result; if we run a single instance of "multiplicative weights" or "follow the perturbed leader", we cannot guarantee non-discrimination with respect to equalized error rates (Theorem 4). We then suggest that the property of not performing better than the best predictor is also crucial; in particular, better algorithms that satisfy the stronger guarantee of low shifting regret [21, 6, 34] are also not able to guarantee this non-discrimination (Theorem 5). These algorithms are considered superior to classical no-regret algorithms as they can better adapt to changes in the environment, which has nice implications in game-theoretic settings [35]. Our latter impossibility result is a first application where having these strong guarantees against changing benchmarks is not necessarily desired and therefore is of independent learning-theoretic interest.

## 1.2 Related work

There is a large line of work on fairness and non-discrimination in machine learning (see [36, 8, 13, 41, 22, 20, 10, 28, 26] for a non-exhaustive list). We elaborate on works that either study group notions of fairness or fairness in online learning.

The last decade has seen a lot of work on group notions of fairness, mostly in classification setting. Examples include notions that compare the percentage of members predicted positive such as demographic parity [8], disparate impact [15], equalized odds [20] and calibration across groups [10, 28]. There is no consensus on a universal fairness notion; rather the specific notion considered is largely task-specific. In fact, previous works identified that these notions are often not compatible to each other [10, 28], posed concerns that they may introduce unintentional discrimination [11], and suggested the need to go beyond such observational criteria via causal reasoning [27, 29]. Prior to our work, group fairness notions have been studied primarily in the batch learning setting with the goal of optimizing a loss function subject to a fairness constraint either in a post-hoc correction framework as proposed by Hardt et al. [20] or more directly during training from batch data [41, 19, 39, 40, 3] which requires care due to the predictors being discriminatory with respect to the particular metric of interest. The setting we focus on in this paper does not have the challenges of the above since all predictors are non-discriminatory; however, we obtain surprising impossibility results due to the ordering in which labels arrive.

Recently fairness in online learning has also started receiving attention. One line of work focuses on imposing a particular fairness guarantee *at all times* for bandits and contextual bandits, either for individual fairness [22, 23] or for group fairness [9]. Another line of work points to counterintuitive externalities of using contextual bandit algorithms agnostic to the group identity and suggest that heterogeneity in data can replace the need for exploration [37, 24]. Moreover, following a seminal paper by Dwork et al. [13], a line of work aims to treat similar people similarly in online settings [33, 18]. Our work distinguishes itself from these directions mainly in the objective, since we require the non-discrimination to happen *in the long-term* instead of at any time; this extends the classical batch definitions of non-discrimination in the online setting. In particular, we only focus on situations where there are enough samples from each population of interest and we do not penalize the algorithm for a few wrong decisions, leading it to be overly pessimistic. Another difference is that previous work focuses either on individual notions of fairness or on i.i.d. inputs, while our work is about non-i.i.d. inputs in group notions of fairness.

## 2 Model

**Online learning protocol with group context.** We consider the classical online learning setting of prediction with expert advice, where a learner needs to make sequential decisions for $T$ rounds by combining the predictions of a finite set $\mathcal{F}$ of $d$ hypotheses (also referred to as *experts*). We denote the outcome space by $\mathcal{Y}$; in binary classification, this corresponds to $\mathcal{Y} = \{+, -\}$. Additionally, we introduce a set of disjoint groups by $\mathcal{G}$ which identifies subsets of the population based on a protected attribute (such as gender, ethnicity, or income).

The *online learning protocol with group context* proceeds in $T$ rounds. Each round $t$ is associated with a group context $g(t) \in \mathcal{G}$ and an outcome $y(t) \in \mathcal{Y}$. We denote the resulting $T$-length time-group-outcome sequence tuple by $\sigma = \{(t, g(t), y(t)) \in \mathbb{N} \times \mathcal{G} \times \mathcal{Y}\}_{t=1}^{T}$. This is a random variable that can depend on the randomness in the generation of the groups and the outcomes. We use the shorthand $\sigma^{1:\tau} = \{(t, g(t), y(t)) \in \mathbb{N} \times \mathcal{G} \times \mathcal{Y}\}_{t=1}^{\tau}$ to denote the subsequence until round $\tau$. The exact protocol for generating these sequences is described below. At round $t = 1, 2, \ldots, T$:

1. An example with group context $g(t) \in \mathcal{G}$ arrives stochastically or is adversarially selected.

2. The learning algorithm or *learner* $\mathcal{L}$ commits to a probability distribution $p^t \in \Delta(d)$ across experts where $p^t_f$ denotes the probability that she follows the advice of expert $f \in \mathcal{F}$ at round $t$. This distribution $p^t$ can be a function of the sequence $\sigma^{1:t-1}$. We call the learner *group-unaware* if she ignores the group context $g(\tau)$ for all $\tau \leq t$ when selecting $p^t$.

3. An adversary $\mathcal{A}$ then selects an outcome $y(t) \in \mathcal{Y}$. The adversary is called *adaptive* if the groups/outcomes at round $t = \tau + 1$ are a function of the realization of $\sigma^{1:\tau}$; otherwise she is called *oblivious*. The adversary always has access to the learning algorithm, but an adaptive adversary additionally has access to the realized $\sigma^{1:t-1}$ and hence also knows $p^t$.

   Simultaneously, each expert $f \in \mathcal{F}$ makes a prediction $\hat{y}^t_f \in \hat{\mathcal{Y}}$, where $\hat{\mathcal{Y}}$ is a generic prediction space; for example, in binary classification, the prediction space could simply be the positive or negative labels: $\hat{\mathcal{Y}} = \{+, -\}$, or the probabilistic score: $\hat{\mathcal{Y}} = [0, 1]$ with $\hat{y}^t_f$ interpreted as the probability the expert $f \in \mathcal{F}$ assigns to the positive label in round $t$, or even an uncalibrated score like the output of a support vector machine: $\hat{\mathcal{Y}} = \mathbb{R}$.

   Let $\ell : \hat{\mathcal{Y}} \times \mathcal{Y} \to [0, 1]$ be the loss function between predictions and outcomes. This leads to a corresponding loss vector $\ell^t \in [0, 1]^d$ where $\ell^t_f = \ell\left(\hat{y}^t_f, y(t)\right)$ denotes the loss the learner incurs if she follows expert $f \in \mathcal{F}$.

4. The learner then observes the entire loss vector $\ell^t$ (full information feedback) and incurs expected loss $\sum_{f \in \mathcal{F}} p^t_f \ell^t_f$. For classification, this feedback is obtained by observing $y(t)$.

In this paper, we consider a setting where all the experts $f \in \mathcal{F}$ are fair in isolation (formalized below). Regarding the group contexts, our main impossibility results (Theorems 1 and 2) assume that the group contexts $g(t)$ arrive stochastically from a fixed distribution, while our positive result (Theorem 3) holds even when they are adversarially selected.

For simplicity of notation, we assume throughout the presentation that the learner's algorithm is producing the distribution $p^t$ of round $t = \tau + 1$ deterministically based on $\sigma^{1:\tau}$ and therefore all our expectations are taken only over $\sigma$ which is the case in most algorithms. Our results extend when the algorithm uses extra randomness to select the distribution.

**Group fairness in online learning.** We now define non-discrimination (or fairness) with respect to a particular evaluation metric $\mathcal{M}$, e.g. in classification, the false negative rate metric (FNR) is the fraction of examples with positive outcome predicted negative incorrectly. For any realization of the time-group-outcome sequence $\sigma$ and any group $g \in \mathcal{G}$, metric $\mathcal{M}$ induces a subset of the population $\mathcal{S}^\sigma_g(\mathcal{M})$ that is relevant to it. For example, in classification, $\mathcal{S}^\sigma_g(FNR) = \{t : g(t) = g, y(t) = +\}$ is the set of positive examples of group $g$. The performance of expert $f \in \mathcal{F}$ on the subpopulation $\mathcal{S}^\sigma_g(\mathcal{M})$ is denoted by $\mathcal{M}^\sigma_f(g) = \frac{1}{|\mathcal{S}^\sigma_g(\mathcal{M})|} \sum_{t \in \mathcal{S}^\sigma_g(\mathcal{M})} \ell^t_f$.

**Definition 1.** *An expert $f \in \mathcal{F}$ is called **fair in isolation with respect to metric** $\mathcal{M}$ if, for every sequence $\sigma$, her performance with respect to $\mathcal{M}$ is the same across groups, i.e. $\mathcal{M}^\sigma_f(g) = \mathcal{M}^\sigma_f(g')$ for all $g, g' \in \mathcal{G}$.*

The learner's performance on this subpopulation is $\mathcal{M}_\mathcal{L}^\sigma(g) = \frac{1}{|\mathcal{S}_g^\sigma(\mathcal{M})|} \sum_{t \in \mathcal{S}_g^\sigma(\mathcal{M})} \sum_{f \in \mathcal{F}} p_f^t \ell_f^t$. To formalize our non-discrimination desiderata, we require the algorithm to have similar expected performance across groups, when given access to fair in isolation predictors. We make the following assumptions to avoid trivial impossibility results due to low-probability events or underrepresented populations. First, we take expectation over sequences generated by the adversary $\mathcal{A}$ (that has access to the learning algorithm $\mathcal{L}$). Second, we require the relevant subpopulations to be, in expectation, *large enough*. Our positive results do not depend on either of these assumptions. More formally:

**Definition 2.** *Consider a set of experts $\mathcal{F}$ such that each expert is fair in isolation with respect to metric $\mathcal{M}$. Learner $\mathcal{L}$ is called $\alpha$-**fair in composition with respect to metric** $\mathcal{M}$ if, for all adversaries that produce $\mathbb{E}_\sigma[\min(|S_g^\sigma(\mathcal{M})|, |S_{g'}^\sigma(\mathcal{M})|)] = \Omega(T)$ for all $g, g'$, it holds that:*

$$|\mathbb{E}_\sigma[\mathcal{M}_\mathcal{L}^\sigma(g)] - \mathbb{E}_\sigma[\mathcal{M}_\mathcal{L}^\sigma(g')]| \le \alpha.$$

We note that, in many settings, we wish to have non-discrimination with respect to multiple metrics simultaneously. For instance, equalized odds requires fairness in composition both with respect to false negative rate and with respect to false positive rate (defined analogously). Since we provide an impossibility result for equalized odds, focusing on only one metric makes the result even stronger.

**Regret notions.** The typical way to evaluate the performance of an algorithm in online learning is via the notion of *regret*. Regret is comparing the performance of the algorithm to the performance of the best expert in hindsight on the realized sequence $\sigma$ as defined below:

$$Reg_T = \sum_{t=1}^T \sum_{f \in \mathcal{F}} p_f^t \ell_f^t - \min_{f^\star \in \mathcal{F}} \sum_{t=1}^T \ell_{f^\star}^t.$$

In the above definition, regret is a random variable depending on the sequence $\sigma$; therefore depending on the randomness in groups/outcomes.

An algorithm satisfies the no-regret property (or Hannan consistency) in our setting if for any losses realizable by the above protocol, the regret is sublinear in the time horizon $T$, i.e. $Reg_T = o(T)$. This property ensures that, as time goes by, the average regret vanishes. Many online learning algorithms, such as multiplicative weights updates satisfy this property with $Reg_T = O(\sqrt{T \log(d)})$.

We focus on the notion of *approximate regret*, which is a relaxation of regret that gives a small multiplicative slack to the algorithm. More formally, $\epsilon$-approximate regret with respect to expert $f^\star \in \mathcal{F}$ is defined as:

$$ApxReg_{\epsilon,T}(f^\star) = \sum_{t=1}^T \sum_{f \in \mathcal{F}} p_f^t \ell_f^t - (1 + \epsilon) \sum_{t=1}^T \ell_{f^\star}^t.$$

We note that typical algorithms guarantee $ApxReg_{\epsilon,T}(f^\star) = O(\ln(d)/\epsilon)$ simultaneously for all experts $f^\star \in \mathcal{F}$. When the time-horizon is known in advance, by setting $\epsilon = \sqrt{\ln(d)/T}$, such a bound implies the aforementioned regret guarantee. In the case when the time horizon is not known, one can also obtain a similar guarantee by adjusting the learning rate of the algorithm appropriately.

Our goal is to develop online learning algorithms that combine fair in isolation experts in order to achieve both vanishing average expected $\epsilon$-approximate regret, i.e. for any fixed $\epsilon > 0$ and $f^\star \in \mathcal{F}$, $\mathbb{E}_\sigma[ApxReg_{\epsilon,T}(f^\star)] = o(T)$, and also non-discrimination with respect to fairness metrics of interest.

## 3   Impossibility results for equalized odds

In this section, we study a popular group fairness notion, equalized odds, in the context of online learning. A natural extension of equalized odds for online settings would require that the false negative rate, i.e. percentage of positive examples predicted incorrectly, is the same across all groups and the same also holds for the false positive rate. We assume that our experts are fair in isolation with respect to both false negative as well as false positive rate. A weaker notion of equalized odds is *equality of opportunity* where the non-discrimination condition is required to be satisfied only for the false negative rate. We first study whether it is possible to achieve the vanishing regret property

while guaranteeing $\alpha$-fairness in composition with respect to false negative rate for arbitrarily small $\alpha$. When the input is i.i.d., this is trivial as we can learn the best expert in $O(\log d)$ rounds and then follow its advice; since the expert is fair in isolation, this will guarantee vanishing non-discrimination.

In contrast, we show that, in a non-i.i.d. online setting, this goal is unachievable. We demonstrate this in phenomenally benign settings where there are just two groups $\mathcal{G} = \{A, B\}$ that come from a fixed distribution and just two experts that are fair in isolation (with respect to false negative rate) even per round – not only ex post. Our first construction (Theorem 1) shows that any no-regret learning algorithm that is group-unaware cannot guarantee fairness in composition, even in instances that are perfectly balanced (each pair of label and group gets $^1/_4$ of the examples) – the only adversarial component is the order in which these examples arrive. This is surprising because such a task is straightforward in the stochastic setting as all hypotheses are non-discriminatory. We then study whether actively using the group identity can correct the aforementioned similarly to how it enables correction against discriminatory predictors [20]. The answer is negative even in this scenario (Theorem 2): if the population is sufficiently not balanced, any no-regret learning algorithm will be unfair in composition with respect to false negative rate even if they are not group-unaware.

**Group-unaware algorithms.** We first present the impossibility result for group-unaware algorithms. In our construction, the adversary is oblivious, there is perfect balance in groups (half of the population corresponds to each group), and perfect balance within group (half of the labels of each group are positive and half negative).

**Theorem 1.** For all $\alpha < ^3/_8$, there exists $\epsilon > 0$ such that any group-unaware algorithm that satisfies $\mathbb{E}_\sigma\big[ApxReg_{\epsilon,T}(f)\big] = o(T)$ for all $f \in \mathcal{F}$ is $\alpha$-unfair in composition with respect to false negative rate even for perfectly balanced sequences.

*Proof sketch.* Consider an instance that consists of two groups $\mathcal{G} = \{A, B\}$, two experts $\mathcal{F} = \{h_n, h_u\}$, and two phases: Phase I and Phase II. Group $A$ is the group we end up discriminating against while group $B$ is boosted by the discrimination with respect to false negative rate. At each round $t$ the groups arrive stochastically with probability $^1/_2$ each, independent of $\sigma^{1:t-1}$.

The experts output a score value in $\hat{\mathcal{Y}} = [0, 1]$, where score $\hat{y}_f^t \in \hat{\mathcal{Y}}$ can be interpreted as the probability that expert $f$ assigns to label being positive in round $t$, i.e. $y(t) = +$. The loss function is the expected probability of error given by $\ell(\hat{y}, y) = \hat{y} \cdot \mathbf{1}\{y = -\} + (1 - \hat{y}) \cdot \mathbf{1}\{y = +\}$. The two experts are very simple: $h_n$ always predicts negative, i.e. $\hat{y}_{h_n}^t = 0$ for all $t$, and $h_u$ is an unbiased expert who, irrespective of the group or the label, makes an inaccurate prediction with probability $\beta = ^1/_4 + \sqrt{\epsilon}$, i.e. $\hat{y}_{h_u}^t = \beta \cdot \mathbf{1}\{y(t) = -\} + (1 - \beta) \cdot \mathbf{1}\{y(t) = +\}$ for all $t$. Both experts are fair in isolation with respect to both false negative and false positive rates: FNR is 100% for $h_n$ and $\beta$ for $h_u$ regardless the group, and FPR is 0% for $h_n$ and $\beta$ for $h_u$, again independent of the group. The instance proceeds in two phases:

1. Phase I lasts for $^T/_2$ rounds. The adversary assigns negative labels on examples with group context $B$ and assigns a label uniformly at random to examples from group $A$.

2. In Phase II, there are two plausible worlds:

    (a) if the expected probability the algorithm assigns to expert $h_u$ in Phase I is at least $\mathbb{E}_\sigma\left[\sum_{t=1}^{T/2} p_{h_u}^t\right] > \sqrt{\epsilon} \cdot T$ then the adversary assigns negative labels for both groups

    (b) else the adversary assigns positive labels to examples with group context $B$ while examples from group $A$ keep receiving positive and negative labels with probability equal to half.

    We will show that for any algorithm with vanishing approximate regret property, i.e. with $ApxReg_{\epsilon,T}(f) = o(T)$, the condition for the first world is never triggered and hence the above sequence is indeed balanced.

We now show why this instance is unfair in composition with respect to false negative rate. The proof involves showing the following two claims, whose proofs we defer to the supplementary material.

1. In Phase I, any $\epsilon$-approximate regret algorithm needs to select the negative expert $h_n$ most of the times to ensure small approximate regret with respect to $h_n$. This means that, in Phase

I (where we encounter half of the positive examples from group $A$ and none from group $B$), the false negative rate of the algorithm is close to $1$.

2. In Phase II, any $\epsilon$-approximate regret algorithm should quickly catch up to ensure small approximate regret with respect to $h_u$ and hence the false negative rate of the algorithm is closer to $\beta$. Since the algorithm is group-unaware, this creates a mismatch between the false negative rate of $B$ (that only receives false negatives in this phase) and $A$ (that has also received many false negatives before). □

**Group-aware algorithms.** We now turn our attention to group-aware algorithms, that can use the group context of the example to select the probability of each expert and provide a similar impossibility result. There are three changes compared to the impossibility result we provided for group-unaware algorithms. First, the adversary is not oblivious but instead is adaptive. Second, we do not have perfect balance across populations but instead require that the minority population arrives with probability $b < 0.49$, while the majority population arrives with probability $1 - b$. Third, the labels are not equally distributed across positive and negative for each population but instead positive labels for one group are at least a $c$ percentage of the total examples of the group for a small $c > 0$. Although the upper bounds on $b$ and $c$ are not optimized, our impossibility result cannot extend to $b = c = 1/2$. Understanding whether one can achieve fairness in composition for some values of $b$ and $c$ is an interesting open question. Our impossibility guarantee is the following:

**Theorem 2.** For any group imbalance $b < 0.49$ and $0 < \alpha < \frac{0.49 - 0.99b}{1-b}$, there exists $\epsilon_0 > 0$ such that for all $0 < \epsilon < \epsilon_0$ any algorithm that satisfies $\mathbb{E}_\sigma\left[ApxReg_{\epsilon,T}(f)\right] = o(T)$ for all $f \in \mathcal{F}$, is $\alpha$-unfair in composition.

*Proof sketch.* The instance has two groups: $\mathcal{G} = \{A, B\}$. Examples with group context $A$ are discriminated against and arrive randomly with probability $b < 1/2$ while examples with group context $B$ are boosted by the discrimination and arrive with the remaining probability $1 - b$. There are again two experts $\mathcal{F} = \{h_n, h_p\}$, which output score values in $\hat{\mathcal{Y}} = [0, 1]$, where $\hat{y}_f^t$ can be interpreted as the probability that expert $f$ assigns to label being $+$ in round $t$. We use the earlier loss function of $\ell(\hat{y}, y) = \hat{y} \cdot \mathbf{1}\{y = -\} + (1 - \hat{y}) \cdot \mathbf{1}\{y = +\}$. The first expert $h_n$ is again pessimistic and always predicts negative, i.e. $\hat{y}_{h_n}^t = 0$, while the other expert $h_p$ is optimistic and always predicts positive, i.e. $\hat{y}_{h_p}^t = 1$. These satisfy fairness in isolation with respect to equalized odds (false negative rate and false positive rate). Let $c = 1/101^2$ denote the percentage of the input that is about positive examples for $A$, ensuring that $|\mathcal{S}_g^\sigma(FNR)| = \Omega(T)$. The instance proceeds in two phases.

1. Phase I lasts $\Theta \cdot T$ rounds for $\Theta = 101c$. The adversary assigns negative labels on examples with group context $B$. For examples with group context $A$, the adversary acts as following:

   - if the algorithm assigns probability on the negative expert below $\gamma(\epsilon) = \frac{99 - 2\epsilon}{100}$, i.e. $p_{h_n}^t(\sigma^{1:t-1}) < \gamma(\epsilon)$, then the adversary assigns negative label.
   - otherwise, the adversary assigns positive labels.

2. In Phase II, there are two plausible worlds:

   (a) the adversary assigns negative labels to both groups if the expected number of times that the algorithm selected the negative expert with probability higher than $\gamma(\epsilon)$ on members of group $A$ is less than $c \cdot b \cdot T$, i.e. $\mathbb{E}_\sigma\left[\mathbf{1}\{t \leq \Theta \cdot T : g(t) = A, p_{h_n}^t \geq \gamma(\epsilon)\}\right] < c \cdot b \cdot T$.
   (b) otherwise she assigns positive labels to examples with group context $B$ and negative labels to examples with group context $A$.

   Note that, as before, the condition for the first world will never be triggered by any no-regret learning algorithm (we elaborate on that below) which ensures that $\mathbb{E}_\sigma |S_A^\sigma(FNR)| \geq c \cdot b \cdot T$.

The proof is based on the following claims, whose proofs are deferred to the supplementary material.

1. In Phase I, any vanishing approximate regret algorithm enters the second world of Phase II.

2. This implies a lower bound on the false negative rate on $A$, i.e. $FNR(A) \geq \gamma(\epsilon) = \frac{99 - 2\epsilon}{100}$.

3. In Phase II, any $\epsilon$-approximate regret algorithm assigns large enough probability to expert $h_p$ for group $B$, implying an upper bound on the false negative rate on $B$, i.e. $FNR(B) \leq 1/2(1-b)$. Therefore this provides a gap in the false negative rates of at least $\alpha$. □

# 4 Fairness in composition with respect to an alternative metric

The negative results of the previous section give rise to a natural question of whether fairness in composition can be achieved for some other fairness metric in an online setting.

We answer this question positively by suggesting the *equalized error rates* metric $EER$ which captures the average loss over the total number of examples (independent of whether this loss comes from false negative or false positive examples). The relevant subset induced by this metric $\mathcal{S}_g^\sigma(EER)$ is the set of all examples coming from group $g \in \mathcal{G}$. We again assume that experts are fair in isolation with respect to equalized error rate and show that a simple scheme where we run separately one instance of multiplicative weights for each group achieves fairness in composition (Theorem 3). The result holds for general loss functions (beyond pure classification) and is robust to the experts only being approximately fair in isolation. A crucial property we use is that multiplicative weights not only does not perform worse than the best expert; it also does not perform better. In fact, this property holds more generally by online learning algorithms with optimal regret guarantees [14].

Interestingly, not all algorithms can achieve fairness in composition even with respect to this refined notion. We provide two algorithm classes where this is unachievable. First, we show that any algorithm (subject to a technical condition satisfied by algorithms such as multiplicative weights and follow the perturbed leader) that ignores the group identity can be unboundedly unfair with respect to equalized error rates (Theorem 4). This suggests that the algorithm needs to actively discriminate based on the groups to achieve fairness with respect to equalized error rates. Second, we show a similar negative statement for any algorithm that satisfies the more involved guarantee of small shifting regret, therefore outperforming the best expert (Theorem 5). This suggests that the algorithm used should be good but not too good. This result is, to the best of our knowledge, a first application where shifting regret may not be desirable which may be of independent interest.

**The positive result.** We run separate instances of multiplicative weights with a fixed learning rate $\eta$, one for each group. More formally, for each pair of expert $f \in \mathcal{F}$ and group $g \in \mathcal{G}$, we initialize weights $w_{f,g}^1 = 1$. At round $t = \{1, 2, \ldots, T\}$, an example with group context $g(t)$ arrives and the learner selects a probability distribution based to the corresponding weights: $p_f^t = \frac{w_{f,g(t)}^t}{\sum_{j \in \mathcal{F}} w_{j,g(t)}^t}$. Then the weights corresponding to group $g(t)$ are updated exponentially: $w_{f,g}^{t+1} = w_{f,g}^t \cdot (1-\eta)^{\ell_f^t \cdot \mathbf{1}\{g(t)=g\}}$.

**Theorem 3.** For any $\alpha > 0$ and any $\epsilon < \alpha$ such that running separate instances of multiplicative weights for each group with learning rate $\eta = \min(\epsilon, \alpha/6)$ guarantees $\alpha$-fairness in composition and $\epsilon$-approximate regret of at most $O(|\mathcal{G}| \log(d)/\epsilon)$.

*Proof sketch.* The proof is based on the property that multiplicative weights performs not only no worse than the best expert in hindsight but also no better. Therefore the average performance of multiplicative weights at each group is approximately equal to the average performance of the best expert in that group. Since the experts are fair in isolation, the average performance of the best expert in all groups is the same which guarantees the equalized error rates desideratum. We make these arguments formal in the supplementary material. □

**Remark 1.** *If the instance is instead only approximately fair in isolation with respect to equalized error rates, i.e. the error rates of the two experts are not exactly equal but within some constant $\kappa$, the same analysis implies $(\alpha + \kappa)$-fairness in composition with respect to equalized error rates.*

**Impossibility results for group-unaware algorithms.** In the previous algorithm, it was crucial that the examples of the one group do not interfere with the decisions of the algorithm on the other group. We show that, had we run one multiplicative weights algorithm in a group-unaware way, we would not have accomplished fairness in composition. In fact, this impossibility result holds for any algorithm with vanishing $\epsilon$-approximate regret where the learning dynamic ($p^t$ at each round $t$) is a deterministic function of the difference between the cumumative losses of the experts (without taking into consideration their identity). This is satisfied, for instance by multiplicative weights and follow the perturbed leader with a constant learning rate. Unlike the previous section, the impossibility results for equalized error rates require groups to arrive adversarially (which also occurs in the above positive result). The proof of the following theorem is provided in the supplementary material.

**Theorem 4.** For any $\alpha > 0$ and for any $\epsilon > 0$, running a single algorithm from the above class in a group-unaware way is $\alpha$-unfair in composition with respect to equalized error rate.

**Impossibility results for shifting algorithms.** The reader may be also wondering whether it suffices to just run separate learning algorithms in the two groups or whether multiplicative weights has a special property. In the following theorem, we show that the latter is the case. In particular, multiplicative weights has the property of not doing better than the best expert in hindsight. The main representative of algorithms that do not have such a property are the algorithms that achieve low approximate regret compared to a shifting benchmark (tracking the best expert). More formally, approximate regret against a shifting comparator $f^\star = (f^\star(1), \ldots, f^\star(T))$ is defined as:

$$ApxReg_{\epsilon,T}(f^\star) = \sum_t p_f^t \ell_f^t - (1 + \epsilon) \sum_t \ell_{f^\star(t)}^t,$$

and typical guarantees are $\mathbb{E}[ApxReg(f^\star)] = O\big(K(f^\star) \cdot \ln(dT)/\epsilon\big)$ where $K(f^\star) = \sum_{t=2}^T \mathbb{1}\{f^\star(t) \neq f^\star(t-1)\}$ is the number of switches in the comparator. We show that any algorithm that can achieve such a guarantee even when $K(f^\star) = 2$ does not satisfy fairness in composition with respect to equalized error rate. This indicates that, for the fairness with equalized error rates purpose, the algorithm not being too good is essential. This is established in the following theorem whose proof is deferred to the supplementary material.

**Theorem 5.** For any $\alpha < \nicefrac{1}{2}$ and $\epsilon > 0$, any algorithm that can achieve the vanishing approximate regret property against shifting comparators $f$ of length $K(f) = 2$, running separate instances of the algorithm for each group is $\alpha$-unfair in composition with respect to equalized error rate.

## 5   Discussion

In this paper, we introduce the study of avoiding discrimination towards protected groups in online settings with non-i.i.d. examples. Our impossibility results for equalized odds consist of only two phases, which highlights the challenge in correcting for historical biases in online decision making.

Our work also opens up a quest towards definitions that are relevant and tractable in non-i.i.d. online settings for specific tasks. We introduce the notion of equalized error rates that can be a useful metric for non-discrimination in settings where all examples that contribute towards the performance also contribute towards fairness. This is the case in settings that all mistakes are similarly costly as is the case in speech recognition, recommender systems, or online advertising. However, we do not claim that its applicability is universal. For instance, consider college admission with two perfectly balanced groups that correspond to ethnicity (equal size of the two groups and equal number of positive and negatives within any group). A racist program organizer can select to admit all students of the one group and decline the students of the other, while satisfying equalized error rates – this does not satisfy equalized odds. Given the impossibility result we established for equalized odds, it is interesting to identify definitions that work well for different tasks one encounters in online non-i.i.d. settings. Moreover, although our positive results extend to the case where predictors are vetted to be approximately non-discriminatory, they do not say anything about the case where the predictors do not satisfy this property. We therefore view our work only as a first step towards understanding non-discrimination in non-i.i.d. online settings.

## Acknowledgements

The authors would like to thank Manish Raghavan for useful discussions that improved the presentation of the paper. This work was supported by the NSF grants CCF-1800317 and CCF-1563714, as well as a Google Ph.D. Fellowship.

## Footnotes

[1]In contrast, in equalized odds, a misprediction only costs to the false negative metric if the label is positive.

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
