[Supplementary Material]

# On preserving non-discrimination when combining expert advice (supplementary material)

**Avrim Blum**
TTI-Chicago
avrim@ttic.edu

**Suriya Gunasekar**
TTI-Chicago
suriya@ttic.edu

**Thodoris Lykouris**
Cornell University
teddlyk@cs.cornell.edu

**Nathan Srebro**
TTI-Chicago
nati@ttic.edu

## A  Supplementary material for proof of Theorem 1

We here provide the proof of the claims omitted in the main text of the paper.

**Upper bound on probability of playing $h_u$ in Phase I.**   We now formalize the first claim by showing that any algorithm with $\mathbb{E}_\sigma\left[\sum_{t=1}^{T/2} p_{h_u}^t\right] > \sqrt{\epsilon}\cdot T$ does not satisfy the approximate regret property. The algorithm suffers an expected loss of $\beta$ every time it selects expert $h_u$. On the other hand, when selecting expert $h_n$, it suffers a loss of $0$ for members of group $B$ and an expected loss of $1/2$ for members of group $A$. As a result, the expected loss of the algorithm in the first phase is:

$$
\begin{aligned}
\mathbb{E}_\sigma\left[\sum_{t=1}^{T/2}\sum_{f\in\mathcal{F}} p_f^t\cdot\ell_f^t\right] &= \mathbb{E}_\sigma\left[\sum_{t=1}^{T/2} p_{h_u}^t\right]\cdot\beta + \mathbb{E}_\sigma\left[\sum_{t=1}^{T/2} p_{h_n}^t\cdot\mathbf{1}_{g(t)=A}\right]\cdot\frac{1}{2}\\
&= \mathbb{E}_\sigma\left[\sum_{t=1}^{T/2} p_{h_u}^t\right]\cdot\beta + \left(\frac{T}{2}-\mathbb{E}_\sigma\left[\sum_{t=1}^{T/2} p_{h_u}^t\right]\right)\cdot\frac{1}{4}\\
&= \frac{T}{8} + \left(\beta-\frac{1}{4}\right)\cdot\mathbb{E}_\sigma\left[\sum_{t=1}^{T/2} p_{h_u}^t\right] = \frac{T}{8} + \sqrt{\epsilon}\cdot\mathbb{E}_\sigma\left[\sum_{t=1}^{T/2} p_{h_u}^t\right]
\end{aligned}
$$

In contrast, the negative expert has, in Phase I, expected loss of:

$$
\mathbb{E}_\sigma\left[\sum_{t=1}^{T/2}\ell_{h_n}^t\right] = \frac{T}{8}.
$$

Therefore, if $\mathbb{E}_\sigma\left[\sum_{t=1}^{T/2} p_{h_u}^t\right] > \sqrt{\epsilon}\cdot T$, the $\epsilon$-approximate regret of the algorithm with respect to $h_n$ is linear to the time-horizon $T$ (and therefore not vanishing) since:

$$
\mathbb{E}_\sigma\left[\sum_{t=1}^{T/2}\sum_{f\in\mathcal{F}} p_f^t\cdot\ell_f^t - (1+\epsilon)\sum_{t=1}^{T/2}\ell_{h_N}^t\right] = \frac{T}{8} + \sqrt{\epsilon}\cdot\mathbb{E}_\sigma\left[\sum_{t=1}^{T/2} p_{h_u}^t\right] - (1+\epsilon)\frac{T}{8} > \frac{7\epsilon}{8}\cdot T.
$$

**Upper bound on probability of playing $h_n$ in Phase II.**   Regarding the second claim, we first show that $\mathbb{E}_\sigma\left[\sum_{t=T/2+1}^{T} p_{h_n}^t\right] \le 16\sqrt{\epsilon}\cdot T$ for any $\epsilon$-approximate regret algorithm with $\epsilon < 1/16$.

The expected loss of the algorithm in the second phase is:

$$
\mathbb{E}_\sigma\left[\sum_{t=T/2+1}^{T}\sum_{f\in\mathcal{F}} p_f^t\ell_f^t\right] = \mathbb{E}_\sigma\left[\sum_{t=T/2+1}^{T} p_{h_n}^t\right]\cdot\frac{3}{4} + \left(\frac{T}{2}-\mathbb{E}_\sigma\left[\sum_{t=T/2+1}^{T} p_{h_n}^t\right]\right)\cdot\beta.
$$

Since, in Phase I, the best case scenario for the algorithm is to always select expert $h_n$ and incur a loss of $T/8$, this implies that for $\epsilon < 1/16$:

$$\mathbb{E}_\sigma \left[ \sum_{t=1}^{T} \sum_{f \in \mathcal{F}} p_f^t \ell_f^t \right] \geq \frac{T}{8} + \frac{T}{2} \cdot \beta + \mathbb{E}_\sigma \left[ \sum_{t=T/2+1}^{T} p_{h_n}^t \right] \cdot \left( \frac{3}{4} - \beta \right)$$

$$= \frac{(1 + 2\sqrt{\epsilon}) \cdot T}{4} + \mathbb{E}_\sigma \left[ \sum_{t=T/2+1}^{T} p_{h_n}^t \right] \cdot \left( \frac{1}{2} - \sqrt{\epsilon} \right)$$

$$> \frac{T}{4} + \mathbb{E}_\sigma \left[ \sum_{t=T/2+1}^{T} p_{h_n}^t \right] \cdot \frac{1}{4}.$$

On the other hand, the cumulative expected loss of the $\beta$-inaccurate expert $h_u$ is

$$\mathbb{E} \left[ \sum_{t=1}^{T} \ell_{h_u}^t \right] = \beta \cdot T = \frac{T}{4} + \sqrt{\epsilon} \cdot T.$$

Therefore, if the algorithm has $\mathbb{E}_\sigma \left[ \sum_{t=T/2+1}^{T} p_{h_n}^t \right] > 16\sqrt{\epsilon} \cdot T$, the $\epsilon$-approximate regret of the algorithm with respect to $h_u$ is linear to the time-horizon since $\epsilon \leq 1$, we have:

$$\mathbb{E}_\sigma \left[ \sum_{t=1}^{T} \sum_{f \in \mathcal{F}} p_f^t \ell_f^t - (1 + \epsilon) \sum_{t=1}^{T} \ell_{h_u}^t \right] > \left( \frac{T}{4} + \mathbb{E}_\sigma \left[ \sum_{t=T/2+1}^{T} p_{h_n}^t \right] \cdot \frac{1}{4} \right) - (1 + \epsilon) \cdot \left( \frac{T}{4} + \sqrt{\epsilon} \cdot T \right)$$

$$\geq \mathbb{E}_\sigma \left[ \sum_{t=T/2+1}^{T} p_{h_n}^t \right] \cdot \frac{1}{4} - 3\sqrt{\epsilon} \cdot T > \sqrt{\epsilon} \cdot T$$

The last inequality holds since $\epsilon \cdot T/4 + \epsilon \cdot \sqrt{\epsilon} \cdot T + \sqrt{\epsilon} \cdot T \leq 3\sqrt{\epsilon} \cdot T$ for $\epsilon \leq 1$.

Thus, we have shown that for, for $\epsilon < 1/16$, any algorithm with vanishing approximate regret, necessarily we have $\mathbb{E}_\sigma \left[ \sum_{t=T/2+1}^{T} p_{h_n}^t \right] \leq 16\sqrt{\epsilon} \cdot T$.

**Gap in false negative rates between groups $A$ and $B$.** We now compute the expected false negative rates for the two groups, assuming that $\epsilon < 1/16$. Since we focus on algorithms that satisfy the vanishing regret property, we have already established that:

$$\mathbb{E}_\sigma \left[ \sum_{t=1}^{T/2} p_{h_u}^t \right] \leq \sqrt{\epsilon} \cdot T \quad \text{and} \quad \mathbb{E}_\sigma \left[ \sum_{t=T/2+1}^{T} p_{h_n}^t \right] \leq 16\sqrt{\epsilon} \cdot T. \tag{1}$$

For ease of notation, let $G_{A,+}^t = \mathbf{1}\{g(t) = A, y(t) = +\}$ and $G_{B,+}^t = \mathbf{1}\{g(t) = B, y(t) = +\}$. Since the group context at round $t$ arrives independent of $\sigma^{1:t-1}$ and the adversary is oblivious, we have that $G_{A,+}^t, G_{B,+}^t$ are independent of $\sigma^{1:t-1}$, and hence also independent of $p_{h_u}^t, p_{h_n}^t$.

Since the algorithm is group-unaware, the expected cumulative probability that the algorithm uses $h_n$ in Phase II is the same for both groups. We combine this with the facts that under the online learning protocol with group context, examples of group $B$ arrive stochastically with probability half but only receive positive labels in Phase II, we obtain:

$$\mathbb{E}_\sigma \left[ \sum_{t=T/2+1}^{T} p_{h_n}^t \cdot G_{B,+}^t \right] = \frac{1}{2} \cdot \mathbb{E}_\sigma \left[ \sum_{t=T/2+1}^{T} p_{h_n}^t \right] \leq 8\sqrt{\epsilon} \cdot T. \tag{2}$$

Recall that group $B$ in Phase I has no positive labels, hence the false negative rate on group $B$ is:

$$\mathbb{E}_\sigma[FNR_\mathcal{L}^\sigma(B)] = \mathbb{E}_\sigma\left[\frac{\sum_{t=T/2+1}^T G_{B,+}^t \cdot \left(p_{h_u}^t \cdot \beta + p_{h_n}^t \cdot 1\right)}{\sum_{t=T/2+1}^T \cdot G_{B,+}^t}\right]$$

$$= \beta + \mathbb{E}_\sigma\left[\frac{(1-\beta) \cdot \sum_{t=T/2+1}^T G_{B,+}^t \cdot p_{h_n}^t}{\sum_{t=T/2+1}^T G_{B,+}^t}\right]$$

In order to upper bound the above false negative rate, we denote the following good event by

$$\mathcal{E}^B(\eta) = \left\{\sigma^{1:T} : \sum_{t=T/2+1}^T G_{B,+}^t > (1-\eta)\,\mathbb{E}\left[\sum_{t=T/2+1}^T G_{B,+}^t\right]\right\}.$$

By Chernoff bound, the probability of the bad event is:

$$\mathbb{P}\left[\neg\mathcal{E}^B(\eta)\right] = \exp\left(-\frac{\eta^2 \cdot \mathbb{E}\left[\sum_{t=T/2+1}^T G_{B,+}^t\right]}{2}\right).$$

For $\eta^B = \sqrt{16\log(T)/T}$, this implies that $\mathbb{P}[\neg\mathcal{E}^B(\eta^B)] \leq 1/T^2$ since $\mathbb{E}_\sigma[\sum_{t=T/2+1}^T G_{B,+}^t] = T/4$.

Therefore, by first using the bound on $\sum_{t=T/2+1}^T G_{B,+}^t$ on the good event and the bound on the probability of the bad event, and then taking the limit $T \to \infty$, it holds that:

$$\mathbb{E}_\sigma[FNR_\mathcal{L}^\sigma(B)] = \beta + \mathbb{E}_\sigma\left[\frac{(1-\beta) \cdot \sum_{t=T/2+1}^T G_{B,+}^t \cdot p_{h_n}^t}{\sum_{t=T/2+1}^T G_{B,+}^t}\right]$$

$$\leq \beta + \frac{1-\beta}{1-\eta^B} \cdot \frac{8\sqrt{\epsilon} \cdot T}{T/4} \cdot \mathbb{P}[\mathcal{E}^B(\eta^B)] + 1 \cdot \mathbb{P}[\neg\mathcal{E}^B(\eta^B)]$$

$$\leq \beta + \frac{32\sqrt{\epsilon}}{1-\eta^B} + \frac{1}{T^2} \to \frac{1}{4} + 33\sqrt{\epsilon}.$$

We now move to the false negative rate of $A$:

$$\mathbb{E}_\sigma[FNR_\mathcal{L}^\sigma(A)] = \mathbb{E}_\sigma\left[\frac{\sum_{t=1}^T G_{A,+}^t \cdot \left(p_{h_u}^t \cdot \beta + p_{h_n}^t \cdot 1\right)}{\sum_{t=1}^T G_{A,+}^t}\right].$$

Similarly as before, letting $\mathcal{E}^A(\eta) = \left\{\sigma^{1:T} : \sum_{t=1}^T G_{A,+}^t < (1+\eta)\,\mathbb{E}\left[\sum_{t=1}^T G_{A,+}^t\right]\right\}$ and, since $\mathbb{P}[\neg\mathcal{E}^A(\eta)] = \exp\left(-\eta^2 \cdot \mathbb{E}[\sum_{t=1}^T G_{A,+}^t]/3\right)$, we obtain that, for $\eta^A = \sqrt{24\log(T)/T}$, $\mathbb{P}[\neg\mathcal{E}^A(\eta^A)] = 1/T^2$.

Recall that for our instance $\mathbb{E}_\sigma\left[G_{A,+}^t\right] = T/4$ and $G_{A,+}^t$ is independent of $p_{h_u}^t$. From our previous analysis we also know that:

$$\mathbb{E}_\sigma\left[\sum_{t=1}^{T/2} p_{h_u}^t G_{A,+}^t\right] \leq \frac{\sqrt{\epsilon} \cdot T}{4} \quad \text{and} \quad \mathbb{E}_\sigma\left[\sum_{t=T/2+1}^T p_{h_u}^t G_{A,+}^t\right] \leq \frac{T}{8} \qquad (3)$$

As a result, using that $\mathbb{E}\left[\sum_{t=1}^{T/2} G_{A,+}^t\right] = \mathbb{E}\left[\sum_{t=T/2+1}^T G_{A,+}^t\right] = \frac{T}{8}$ and Inequalities (3), we obtain:

$$\mathbb{E}_\sigma\left[\sum_{t=1}^T G_{A,+}^t \cdot \left(p_{h_u}^t \cdot \beta + p_{h_n}^t \cdot 1\right)\right] = \mathbb{E}_\sigma\left[\sum_{t=1}^T G_{A,+}^t \cdot - \sum_{t=1}^T G_{A,+}^t \cdot p_{h_u}^t(1-\beta)\right]$$

$$\geq \frac{T}{4}\left(1 - (1-\beta) \cdot (\frac{1}{2} + \sqrt{\epsilon})\right).$$

Therefore, similarly with before, it holds that:

$$\mathbb{E}_\sigma[FNR_\mathcal{L}^\sigma(A)] = \mathbb{E}_\sigma\left[\frac{\sum_{t=1}^T G_{A,+}^t \cdot \left(p_{h_u}^t \cdot \beta + p_{h_n}^t \cdot 1\right)}{\sum_{t=1}^T G_{A,+}^t}\right]$$

$$\geq \frac{1 - (1-\beta) \cdot (\frac{1}{2} + \sqrt{\epsilon})}{(1+\eta^A)} \cdot \mathbb{P}\left[\mathcal{E}^A(\eta^A)\right] + 0 \cdot \mathbb{P}\left[\neg\mathcal{E}^A(\eta^A)\right]$$

$$\geq \frac{\frac{1}{2} - \sqrt{\epsilon} + \beta/2}{1+\eta^A}\left(1 - \frac{1}{T^2}\right) > \frac{\frac{1}{2} - \sqrt{\epsilon} + 1/8}{1+\eta^A}\left(1 - \frac{1}{T^2}\right) \to \frac{5}{8} - \sqrt{\epsilon}.$$

As a result, the difference between the false negative rate in group $A$ and the one at group $B$ is $3/8 + 34\sqrt{\epsilon}$ which can go arbitrarily close to $3/8$ by appropriately selecting $\epsilon$ to be small enough, for any vanishing approximate regret algorithm. This concludes the proof.

## B   Supplementary material for proof of Theorem 2

We here provide the proof of the claims omitted in the main text of the paper.

**Proof of first claim.**   To prove the first claim, we apply the method of contradiction. Assume that the algorithm has $\mathbb{E}_\sigma\left[\mathbf{1}\{t \leq \Theta \cdot T : g(t) = A, p_{h_n}^t \geq \gamma(\epsilon)\}\right] < c \cdot b \cdot T$. This means that the algorithm faces an expectation of at least $(\Theta - c) \cdot b \cdot T$ negative examples, while predicting the negative expert with probability at most $\gamma(\epsilon) = \frac{99-2\epsilon}{100}$, thereby making an error with probability $1 - \gamma(\epsilon)$. Therefore the expected loss of the algorithm is at least:

$$\mathbb{E}_\sigma\left[\sum_{t=1}^{\Theta \cdot T}\sum_{f\in\mathcal{F}} p_f^t \cdot \ell_f^t\right] > (\Theta - c) \cdot b \cdot T \cdot (1 - \gamma(\epsilon)) = c \cdot b \cdot (1+2\epsilon) \cdot T.$$

At the same time, expert $h_n$ makes at most $c \cdot b \cdot T$ errors (at the positive examples)

$$\mathbb{E}_\sigma\left[\sum_{t=1}^T \ell_{h_n}^t\right] < c \cdot b \cdot T.$$

Therefore, if $\mathbb{E}_\sigma\left[\mathbf{1}\{t \leq \Theta \cdot T : g(t) = A, p_{h_n}^t \geq f(\epsilon)\}\right] < c \cdot b \cdot T$, the $\epsilon$-approximate regret of the algorithm with respect to $h_n$ is linear to the time-horizon $T$ (and therefore not vanishing) since:

$$\mathbb{E}_\sigma\left[\sum_{t=1}^T\sum_{f\in\mathcal{F}} p_f^t \ell_f^t - (1+\epsilon)\sum_{t=1}^T \ell_{h_n}^t\right] > \epsilon \cdot c \cdot b \cdot T.$$

This violates the vanishing approximate regret property, thereby leading to contradiction.

**Proof of second claim.**   The second claim follows directly by the above construction, since positive examples only appear in Phase I when the probability of error on them is greater than $\gamma(\epsilon)$.

**Proof of third claim.**   Having established that any vanishing approximate regret algorithm will always enter the second world, we now focus on the expected loss of expert $h_p$ in this case. This expert makes errors at most in all Phase I and in the examples of Phase II with group context $A$:

$$\mathbb{E}_\sigma\left[\sum_{t=1}^T \ell_{h_p}^t\right] \leq \Theta \cdot T + b \cdot (1-\Theta) \cdot T \leq \Theta \cdot T + 0.49 \cdot (1-\Theta) \cdot T$$

Since group $B$ has only positive examples in Phase II, the expected loss of the algorithm is at least:

$$\mathbb{E}_\sigma\left[\sum_{t=1}^T\sum_{f\in\mathcal{F}} p_f^t \ell_f^t\right] \geq \mathbb{E}_\sigma\left[\sum_{t=\Theta\cdot T+1}^T p_{h_n}^t \cdot \mathbf{1}_{g(t)=B}\right]$$

We now show that $\mathbb{E}_\sigma\left[\sum_{t=\Theta\cdot T+1}^{T} p_{h_n}^t \cdot \mathbf{1}_{g(t)=B}\right] \le (1/2 + \epsilon) \cdot (1 - \Theta) \cdot T$. If this is not the case, then the algorithm does not have vanishing $\epsilon$-approximate regret with respect to expert $h_p$ since:

$$\mathbb{E}_\sigma\left[\sum_{t=1}^{T}\sum_{f\in\mathcal{F}} p_f^t \ell_f^t - (1+\epsilon)\sum_{t=1}^{T}\ell_{h_p}^t\right] > \left(\frac{1}{2}+\epsilon\right)\cdot(1-\Theta)T - (1+\epsilon)\cdot 0.49\cdot(1-\Theta)T - (1+\epsilon)\Theta T$$

$$\ge \left(\frac{1}{2}+\epsilon-0.49-0.49\epsilon\right)\cdot(1-\Theta)\cdot T - (1+\epsilon)\cdot\Theta\cdot T$$

$$> (0.01+0.51\epsilon)\cdot\frac{100}{101}\cdot T - \frac{1+\epsilon}{101}\cdot T \ge \frac{50}{101}\epsilon\cdot T$$

Given the above, we now establish a gap in the fairness with respect to false negative rate. Since group $A$ only experiences positive examples when expert $h_n$ is offered probability higher than $\gamma(\epsilon) = \frac{99-2\epsilon}{100}$, this means that:

$$\mathbb{E}_\sigma[FNR_{\mathcal{L}}^\sigma(A)] \to 0.99 - 0.02\epsilon$$

Regarding group $B$, we need to take into account the low-probability event that the actual realization has significantly less than $(1-b)(1-\Theta)\cdot T$ examples of group $B$ in Phase II (all are positive examples). This can be handled via similar Chernoff bounds as in the proof of the previous theorem. As a result, the expected false negative rate at group $B$ is:

$$\mathbb{E}_\sigma[FNR_{\mathcal{L}}^\sigma(B)] \to \frac{\mathbb{E}_\sigma\left[\sum_{t=\Theta\cdot T+1}^{T} p_{h_n}^t \cdot \mathbf{1}_{g(t)=B}\right]}{\mathbb{E}_\sigma\left[\sum_{t=\Theta\cdot T+1}^{T} \mathbf{1}_{g(t)=B}\right]} \le \frac{(1/2+\epsilon)\cdot(1-\Theta)\cdot T}{(1-b)\cdot(1-\Theta)\cdot T} = \frac{1/2+\epsilon}{1-b}$$

which establishes a gap in the fairness with respect to false negative rate of $\alpha \to \frac{0.49-0.99b}{1-b}$ selecting $\epsilon > 0$ appropriately small.

## C  Supplementary material for proof of Theorem 3

We here provide the details of the proof of Theorem 3.

We follow the classical potential function analysis of multiplicative weights but apply bidirectional bounds to also lower bound the performance of the algorithm by the performance of the comparator. For each group $g \in \mathcal{G}$ and every expert $f \in \mathcal{F}$, let $L_{f,g} = \sum_{t:g(t)=g} \ell_f^t \cdot \mathbf{1}\{g(t) = g\}$ be the cumulative loss of expert $f$ in examples with group context $g$, and $\hat{L}_g = \sum_{t=1}^{T}\sum_{f\in\mathcal{F}} p_f^t \ell_f^t \cdot \mathbf{1}\{g(t) = g\}$ to denote the expected loss of the algorithm on these examples. We also denote the best in hindsight expert on these examples by $f^\star(g) = \arg\min_{f\in\mathcal{F}} L_{f,g}$. Recall that $w_{f,g}^t = (1-\eta)^{\sum_{\tau \le t: g(\tau)=g} \ell_f^\tau}$ is the weight of expert $f$ in the instance of group $g$ and let $W_{t,g} = \sum_{j\in\mathcal{F}} w_{j,g}^t$ be its potential function.

To show that the algorithm does not perform much worse than any expert, we follow the classical potential function analysis and, since $(1-\eta)^x \le 1 - \eta x$ for all $x \in [0,1]$ and $\eta \le 1$, we obtain:

$$W_{t+1,g} = \sum_{j\in\mathcal{F}} w_{j,g}^t \cdot (1-\eta)^{\ell_j^t\cdot\mathbf{1}\{g(t)=g\}} \le \sum_{j\in\mathcal{F}} w_{j,g}^t \cdot (1 - \eta\ell_j^t \cdot \mathbf{1}\{g(t)=g\})$$

$$= W_{t,g}\cdot\left(1 - \eta\sum_{j\in\mathcal{F}} p_j^t \ell_j^t\right).$$

By the classical analysis, for all $f \in \mathcal{F}$ and $g \in \mathcal{G}$:

$$w_{f,g}^{T+1} = (1-\eta)^{\sum_{t=1}^{T} \ell_f^t\cdot\mathbf{1}\{g^t=g\}} \le W_{T+1,g} \le d\cdot\prod_{t=1}^{T}\left(1 - \eta\sum_{j\in\mathcal{F}} p_j^t \ell_j^t \cdot \mathbf{1}\{g(t)=g\}\right)$$

where the left inequality follows from the fact that all summands of $W_{T+1,g}$ are positive and the right inequality follows by unrolling $W_{T+1,g}$ and using that $W_{1,g} = d$.

Taking logarithms and using that $-\eta - \eta^2 < \ln(1-\eta) < -\eta$ for $\eta < 1/2$, this implies that for all $f \in \mathcal{F}$:

$$\hat{L}_g \le (1+\eta)\cdot L_{f,g} + \frac{\ln(d)}{\eta} \tag{4}$$

We now use the converse side of the inequalities to show that multiplicative weights also does not perform much better than the best expert in hindsight $f^\star(g)$. Using that $(1 - \eta)^x \geq 1 - \eta(1 + \eta)x$ for all $x \in [0, 1]$, we obtain:

$$W_{t+1,g} = \sum_{j \in \mathcal{F}} w_{j,g}^t \cdot (1 - \eta)^{\ell_j^t \cdot \mathbf{1}\{g(t)=g\}} \geq \sum_{j \in \mathcal{F}} w_{j,g}^t \cdot \left(1 - \eta(1+\eta) \cdot \ell_j^t \cdot \mathbf{1}\{g(t) = g\}\right)$$

$$= W_{t,g} \cdot \left(1 - \eta(1+\eta) \sum_{j \in \mathcal{F}} p_i^t \ell_i^t\right).$$

Using that $f^\star(g)$ is the best expert in hindsight, we can upper bound $\sum_{j \in \mathcal{F}} w_{j,g}^t \leq d \cdot \max_{j \in \mathcal{F}} w_{j,g}^t = d \cdot \max_{f \in \mathcal{F}} (1 - \eta)^{\sum_{t=1}^t \ell_f^t \mathbf{1}\{g^t = g\}}$. Similarly to before, it therefore follows that:

$$d \cdot (1 - \eta)^{\sum_{t=1}^T \ell_{f^\star(g)}^t \mathbf{1}\{g^t=g\}} \geq W_{T+1} \geq d \cdot \prod_{t=1}^T \left(1 - \eta(1+\eta) \sum_{j \in \mathcal{F}} p_j^t \ell_j^t\right)$$

which, for $\eta < 1/2$, implies that:

$$\widehat{L}_g \geq (1 - 4\eta) \cdot L_{f^\star(g),g} \tag{5}$$

The expected $\epsilon$-approximate regret of this algorithm is at most $6 \cdot |\mathcal{G}|$ times the one of a single multiplicative weights instance (by summing over inequalities (4) for all $g \in \mathcal{G}$ and since $\epsilon/6 \leq \eta \leq \epsilon$). What is left to show is that the $\alpha$-fairness in composition guarantee is satisfied, that is there exists $T_0$ (function of $\alpha$ and $\epsilon$) such that when the number of examples from each group is at least $T_0$, the maximum difference between average expected losses across groups is bounded by $\alpha$. Let $g^\star$ be the group with the smallest average expected loss. We will show that the maximum difference from the average expected loss of any other group $g$ is at most $\alpha$ for $T_0 = 6\ln(d)/\eta\alpha$. Since the experts are fair in isolation, we know that $\frac{L_{f,g}}{|\{t:g^t=g\}|} = \frac{L_{f,g'}}{|\{t:g^t=g'\}|}$ for all $f \in \mathcal{F}$ and $g, g' \in \mathcal{G}$. Combining this with inequalities (4) and (5) and the fact that the losses are in $[0, 1]$ and $\eta \leq \alpha/6$, we obtain:

$$\frac{\widehat{L}_g}{|\{t : g(t) = g\}|} - \frac{\widehat{L}_{g^\star}}{|\{t : g(t) = g^\star\}|} \leq \frac{(1+\eta) \cdot L_{f^\star(g),g}}{|\{t : g(t) = g\}|} + \frac{\ln(d)}{\eta \cdot |\{t : g(t) = g\}|} - \frac{(1-4\eta) \cdot L_{f^\star(g^\star),g^\star}}{|\{t : g(t) = g^\star\}|}$$

$$\leq 5\eta \cdot \frac{L_{f^\star(g^\star),g^\star}}{|\{t : g(t) = g^\star\}|} + \frac{\ln(d)}{\eta \cdot T_0} \leq \alpha.$$

## D  Proof of Theorem 4

*Proof.* The instance has two groups $\mathcal{G} = \{A, B\}$ that come in an adversarial order, two experts $\mathcal{F} = \{f_1, f_2\}$, and consists of four phases of equal size. At each phase one predictor is always correct and the other one always incorrect.

1. In Phase I $\{1, \ldots, T/4\}$, examples of group $A$ arrive and the first predictor is correct: $\ell_{f_1}^t = 0$ and $\ell_{f_2}^t = 1$.

2. In Phase II $\{T/4 + 1, \ldots, T/2\}$, examples of group $B$ arrive and the second predictor is the correct one: $\ell_{f_1}^t = 1$ and $\ell_{f_2}^t = 0$.

3. In Phase III $\{T/2, \ldots, 3T/4 + 1\}$, examples of group $A$ arrive and the second predictor is again the better one, i.e. $\ell_1^t = 1$ and $\ell_2^t = 0$.

4. Finally, in Phase IV $\{3T/4 + 1, \ldots, T\}$, examples of group $B$ arrive and now the first predictor is accurate: $\ell_1^t = 0$ and $\ell_2^t = 1$.

Note that both experts are fair in isolation with respect to equalized error rates as they both have $50\%$ error rate in each group.

Since the loss of the first expert is 0 in the first quarter of the setting: $\sum_{t=1}^{T/4} \ell_{f_1} = 0$, any algorithm with vanishing approximate regret needs to have sublinear loss during this quarter to be robust against an adversary that continues giving 0 losses to $f_1$. Therefore, in particular, it holds that:

$$\sum_{t=1}^{T/4} p_{f_2} < \frac{1-\alpha}{8} T.$$

As a result, the error rate on group $A$ is at most $EER(A) \leq \frac{1-\alpha}{2}$ in Phase I and, since the algorithm's distribution is deterministic based on the difference in the losses, this also applies to Phase III.

Regarding group $B$, note that any time $t$ where an example of group $B$ arrives has a $1-1$ mapping to the time $t - T/4$ where a member from group $A$ came, where the predictions of the algorithm are the same since the difference between losses are the same. Therefore, by our assumption on the dynamic, the cumulative probability the correct expert is upper bounded by $\frac{1-\alpha}{2}$ which implies that group $B$ incurs an equalized error rate of $EER(B) \geq \frac{1+\alpha}{2}$. Thi concludes the proof. □

## E    Proof of Theorem 5

*Proof.* Our instance has two groups $\mathcal{G} = \{A, B\}$, two experts $\mathcal{F} = \{f_1, f_2\}$, and three phases.

1. Phase I lasts for half of the time horizon $\{1, \ldots, T/2\}$ and during this time, we receive examples from group $A$. At round $t$, the adversary selects loss $\ell_f^t = 1$ for the expert $f \in \mathcal{F}$ that is predicted with higher probability ($p_f^t \leq 1/2$) and $\ell_h^t = 0$ for the other expert $h \in \mathcal{F} - \{f\}$.

2. Phase II lasts $\sum_{\tau=1}^{T/2} \ell_{f_1}^\tau$ rounds and the adversary selects losses $\ell_{f_1}^t = 1$ and $\ell_{f_2}^t = 0$.

3. Phase III lasts $\sum_{\tau=1}^{T/2} \ell_{f_2}^\tau$ rounds and the adversary selects losses $\ell_{f_1}^t = 0$ and $\ell_{f_2}^t = 1$.

Note that the instance is fair in isolation with respect to equalized error rates as the cardinality of both groups is the same (half of the population in each group) and the experts make the same number of mistakes in both groups.

By construction, the algorithm has expected average loss of at least $1/2$ in members of group $A$.

We now focus on group $B$. By the shifting approximate regret guarantee and given that there exists a sequence of experts of length 2 that has 0 loss, it holds that the total loss of the algorithm needs to be sublinear on $T$ and, in particular, at most $(1/2 - \alpha) \cdot \frac{T}{2}$, which implies an expected error rate of $1/2 - \alpha$. Subtracting the two error rates concludes the proof. □