[Reviews · NeurIPS 2018]

Reviewer 1



This paper studies the design of an algorithm for group fairness in online learning. This setup is more realistic than online learning for individual fairness and batch group fairness. The paper is very well written and the contributions are clear. However, I think the assumption that all the predictors are individually non-discriminatory comes a little strong. Although it is fine to use this strong assumption for the impossibility results, I did not find the other two results (the positive ones about the same error rates) practically useful. I think the authors could replace this assumption by making a weaker adversary. For example, assuming that labels come from a distribution instead of adversely getting selected. I think adding more intuition about these two assumptions would help a lot. 1) Assuming every predictor is individually non-discriminatory. 2) Assuming the adversary can adaptively choose the labels.

Reviewer 2



The paper theoretically studies the suitability of achieving a particular definition of fairness, equalized odds (which relates to the false positive rate), in the context of online learning with experts advise (Cesa-Bianchi et al. 2006). In particular, the authors show that achieving an online algorithm that jointly satisfies zero-regret and equalized odds is not possible. Afterward, they show that this is not the case when considering fairness in terms of the total number of errors per group. They also discuss that unfortunately this definition of fairness (also previously discussed in Zafar et al., 2017) is not realistic (or even fair) in many real-world scenarios. In the positive side, I believe that (im)possibility theoretical studies on when a fairness definition can be accomplished is definitely a major contribution to the field. However, I also believe that the paper has important gaps to be filled: 1) Their definition of online learning comes from the game theory literature and does not corresponds to the standard ML view on online learning. However, the authors do not clarify this particular setting in the abstract (neither the title of the paper) and provide any reference to the considered "benign setting"--where at time t there is a set of expert providing their advise about the decision to be made, and the learner select one expert advise (decision) with a fixed probability i~p^t, getting a loss l(t,i) that depends on the selected expert. Is this the actual setting? Please clarify this point in the paper, and add the necessary references. 2) Although the paper is presented in the context of fairness, which is definitely a real problem, the authors do not provide a single real-world example where their setting (based on the game theoretical game benign) would fit. As a consequence, it is hard to evaluate the potential impact of the presented theoretical results in the field. In summary, I believe although the paper presents good ideas and results, it does not provide the necessary context and details to judge the contribution of the paper. Cesa-Bianchi, Nicolo, and Gábor Lugosi. Prediction, learning, and games. Cambridge university press, 2006. B. Zafar, I. Valera, M. Gomez-Rodriguez and K. Gummadi, "Fairness Beyond Disparate Treatment & Disparate Impact: Learning Classification without Disparate Mistreatment", (Full paper) 26th International World Wide Web Conference (WWW), Perth (Australia), April 2017.

Reviewer 3



The authors consider the setting of fairness in online learning. It is one of the initial works in the domain. The ideas used are not very novel but it is interesting as it is one of the first works in the domain. The authors prove an impossibility result to achieve fairness with equal false positive metric but show a positive result with equal expected error metric. Strong Points: [S1] The paper is well written and easy to follow [S2] The authors have done a good job to explain the ideas with simple examples Weak Points [W1] I think the motivation behind equal error rate applications (speecch processing, recommender systems) is not well explained. I would have liked one case study or a small motivational example to better convey the intuitions. [W2] There are no experiments to show the validity of results on real world datasets. It will help us better understand the advantages and disadvantages of the two metrics considered in the paper. Overall, I like the work. It does not involve many fancy results but presents all the results in a simple manner which are easy to follow. I am disappointed to not see any real world experiments though and hence rate it marginally above the bar (explanation in W2).